# Automated Few-shot Classification with Instruction-Finetuned Language Models

**Rami Aly[1*], Xingjian Shi[2†], Kaixiang Lin[3], Aston Zhang[3], Andrew Gordon Wilson[3,4]**
[1]University of Cambridge    [2]Boson AI
[3]Amazon Web Services    [4]New York University
rami.aly@cl.cam.ac.uk, xshiab@connect.ust.hk,
{kaixianl,astonz}@amazon.com, andrewgw@cims.nyu.edu

## Abstract

A particularly successful class of approaches for few-shot learning combines language models with *prompts* – hand-crafted task descriptions that complement data samples. However, designing prompts by hand for each task commonly requires domain knowledge and substantial guesswork. We observe, in the context of classification tasks, that *instruction finetuned* language models are remarkably robust towards some dimensions of a prompt's design. We subsequently propose a simple method to eliminate the need for handcrafted prompts, named AuT-Few. This approach consists of (i) a prompt retrieval module that selects suitable task instructions from the instruction-tuning knowledge base, and (ii) the generation of two distinct, semantically meaningful, class descriptions and a selection mechanism via cross-validation. Over 12 datasets, spanning 8 classification tasks, we show that AuT-Few outperforms current state-of-the-art few-shot learning methods. Moreover, AuT-Few is the best ranking method across datasets on the RAFT few-shot benchmark. Notably, these results are achieved without task-specific handcrafted prompts on unseen tasks.

## 1 Introduction

Collecting annotated data is time-consuming and expensive. The goal of *few-shot learning* is to address this limitation by developing models that generalize from a small number of training examples.

A now dominant paradigm in few-shot learning involves pre-training a large language model (PLM) on unsupervised language modelling objectives, combined with supervised fine-tuning (Kaplan et al., 2020; Wei et al., 2022b). Fine-tuning on a *variety* of classification tasks improves generalization to new unseen tasks even further (Sanh et al., 2022; Wei et al., 2022b; Chung et al., 2022).

*Prompts*, instructions that describe the tasks in natural language, are crucial to successful fine-tuning on many tasks. Typically, prompts consist of two components: *task templates* and *answer choices*. Task templates are textual instructions about the task. Answer choices are semantic descriptions of the categorical labels. Supervised training on prompted samples, as shown in Figure 1, helps PLMs generalize when instructed via prompts on a new problem (here *natural language inference*). Following Lin et al. (2022), we use the term *upstream model* for these instruction-finetuned PLMs. These prompted upstream models provide state-of-the-art few-shot learning ( (Liu et al., 2022), yet they still rely on strenuous manual intervention from manually crafted prompts, designed by experts with domain knowledge about the underlying tasks.

| | **Task Template** | **Answer Choices** |
|---|---|---|
| **Instruction Tuning Prompts** | | |
| **Sentiment** | Review: We came here on Saturday night [...] How does the reviewer feel about the movie? | 0: very negative [...] 5: very positive |
| **Paraph.** | Last year, Comcast [...] Is that a paraphrase of the sentence Comcast has about [...]. | 0: Yes 1: No |
| **Unseen Target Task Prompts** | | |
| **NLI** | Given Oil prices fall back as Yukos oil threat lifted. can we guarantee that Oil prices rise. is true? | 0: Yes 1: No |

Figure 1: Instruction-tuning uses prompts to specify the task via templates (blue) and label descriptions via answer choices (magenta). Fine-tuning on multiple instructed tasks improves generalization to new ones.

In this paper, we are concerned with an *automated few-shot classification* regime, where the algorithm can only access the training samples

---

[*]Work done while interning at Amazon Web Services.
[†]Work done when author was working at Amazon Web Services.

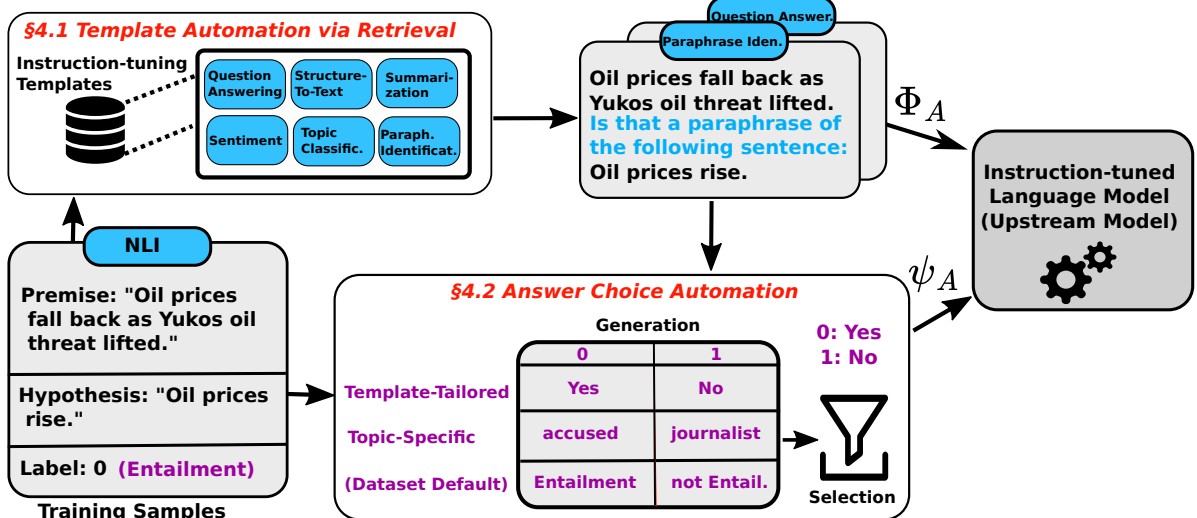

Figure 2: A schematic view of our prompt automation method, AuT-Few, consisting of: the retrieval of templates from the instruction tuning collection (§4.1), and the generation of template-tailored and topic-specific answer choices and the configuration amongst them (and optionally the default dataset label text) (§4.2).

and their categorical labels. While efforts have been made to automate prompting, these methods are not directly transferable to upstream models. Most techniques target prompted masked language models (i.e. encoder-only models, that make predictions over continuous embeddings via its mask token (Gao et al., 2021, *inter alia*). Automation methods for models with a discrete output space (i.e. a decoder over the vocabulary) are costly and limited to the automation of the task template, still relying on handcrafted descriptions of labels (Liu et al., 2021; Zhou et al., 2023).

To automate few-shot learning with upstream models, we analyse the role of prompts across various classification tasks and we observe that upstream models exhibit low variability towards task-unspecific templates. In contrast, the selection of suitable answer choices can be important, yet answer choices do not need to be tailored to the specific instruction (e.g. *Yes/No* for a polar question). These insights confirm observations by Webson and Pavlick (2022) in a broader context and they motivate a simple few-shot learning automation method for upstream models, named *AuT-Few*.

AuT-Few builds on the state-of-the-art learning method T-Few (Liu et al., 2022), but crucially *does not use any task-specific handcrafted prompts*. AuT-Few automatically finds the most relevant templates to our target task from the collection prompts used to instruction-tune the upstream model. As illustrated in Figure 2, given an NLI task, AuT-Few might retrieve templates written for paraphrase

identification. To automate answer choices, AuT-Few generates label descriptions tailored to the retrieved templates (e.g., *Yes/No* for a polar question, as for the illustrated paraphrase identification template) and descriptions that capture a class' overall topic (e.g. *Enron/purchase* for Enron spam classification). AuT-Few selects the most appropriate configuration via cross-validation.

AuT-Few outperforms strong baselines, including T-Few (Liu et al., 2022), by 2.1 points over a total of 12 datasets, spanning 8 tasks, *without any task-specific handcrafted prompts*. All but one task are unseen to the upstream models, indicating AuT-Few's strong generalization capabilities. Moreover, by applying AuT-Few to a small upstream model (BART0 (Lin et al., 2022)), we achieve competitive performance and efficiency to the current state-of-the-art prompt-free method, SetFit (Tunstall et al., 2022). Furthermore, AuT-Few achieves the best average rank across datasets on the few-shot RAFT benchmark (Alex et al., 2021). An ablation justifies the components of our automation method.[1]

## 2 Background and Related Work

### 2.1 Instruction-Finetuned Language Models

A language model is instruction-finetuned on prompted samples $D^{\text{src}}$ from various tasks, such as summarization or question answering, by autoregressively generating the target answer choice through standard maximum likelihood training. In-

---

[1]Code at: `https://github.com/Raldir/AuT-Few`.

struction tuning not only improves generalization for large decoder-only models (Wei et al., 2022a), but also for comparably smaller encoder-decoder models, like T0 (Sanh et al., 2022) or BART0 (Lin et al., 2022). Prompt knowledge bases (KB), like PromptSource (Bach et al., 2022), contain prompt instructions for hundreds of tasks. Flan-T5 (Chung et al., 2022) is an improved upstream model scaled to thousands of tasks (Wang et al., 2022b).

**Inference.** We are interested in using upstream models for an unseen few-shot binary or multi-class classification task $D_{\text{test}}^{tgt}$. A prediction $\hat{y}$ with an upstream model $\theta$ is made by computing the length-normalized log probabilities for each class $y \in \mathcal{Y}$, conditioned on the sample $x$, a handcrafted template $\phi_j \in \Phi$ (i.e. task description and sample input formatting), and on the associated answer choices $\psi_j \in \Psi$ (textual descriptions of labels):

$$\text{argmax}_y(\frac{1}{T} \sum_t \log p_\theta(\psi_j(y) \mid x, \phi_j, \psi_j(y)_{<t}),$$

with $T$ being the length of the answer choice of $y$. Since the use of a single prompt might model the expectation over all possible prompts poorly, most systems handcraft multiple prompts for a target task. The expectation is then modelled by randomly drawing a template and its answer choices.

**Parameter-Efficient Finetuning.** Adapting upstream models to a new task or domain on a few available samples $D_{\text{train}}^{tgt}$ via full model finetuning is often infeasible as these models consist of billions of parameters. Parameter-efficient finetuning adds or updates only a small subset of parameters $\theta_{PEFT} \ll \theta$, and largely retains the fine-tuning performance (Karimi Mahabadi et al., 2021; Zhang et al., 2021; Chen et al., 2023). Liu et al. (2022) proposed T-Few and showed that parameter-efficient finetuning an upstream model with T-Few performs better than in-context learning with GPT-3 in the few-shot learning setting. T-Few learns attention and activation re-scaling vectors by optimizing the maximum likelihood estimation and complements it with an unlikelihood loss.

## 2.2 Prompt Automation

**Template Automation.** To automate the instructions as input to the model, previous work uses soft representation in the input via prompt tuning (Liu et al., 2021; Hambardzumyan et al., 2021), generates discrete instructions (Shin et al., 2020;

Gao et al., 2021; Zhou et al., 2023), or combines both via semi-parametric prompt tuning (Bari et al., 2022). However, prompt tuning is brittle to optimize (Hu et al., 2022a; Liu et al., 2022), and the generation of discrete instructions requires substantial computational resources, a particular concern with upstream models as they typically have billions of parameters. The retrieval of instructions is limited to the retrieval of trained soft prompts and samples (Ye et al., 2022), prompt initialization (Vu et al., 2022), or the retrieval of multiple prompt mixtures (Qin and Eisner, 2021; Asai et al., 2022).

**Answer Choice Automation.** Methods to automate label representations are targeting BERT-like masked language models (Devlin et al., 2019), which enables optimization of the output descriptions on continuous vector representation. Shin et al. (2020) train a logistic classifier on embeddings to score tokens in the vocabulary by how well they predict the task labels. Gao et al. (2021) compute the probability for a token to be the masked classification token, by computing the dot product between both embeddings. Wang et al. (2022a) additionally ensure that label tokens belong only to a single class. Alternatively to such discrete search is learning soft output representations of labels via gradient descent (Hambardzumyan et al., 2021; Cui et al., 2022; Hu et al., 2022b; Karimi Mahabadi et al., 2022), or combining both (Ma et al., 2022). Tunstall et al. (2022) propose a fully prompt-free method using Sentence Transformers (Reimers and Gurevych, 2019).

**Novelty.** Prior works on prompt automation are computationally intensive, brittle to optimize, or assume a continuous output representation for each token. By contrast, our proposed approach automates prompts for upstream models, which operate over a discrete output space. We do not insert any additional trainable parameters for automating templates. Instead, our work is the first to use retrieved instruction-finetuning templates for an unseen task directly and to use them to optimize the answer choices via the generation of distinct, semantically meaningful, answer choice configurations.

## 3 How Much Does the Design of Prompts Matter for Upstream Models?

To automate prompts, we need to understand their role in few-shot classification. While previous research suggests that the wording of instructions for

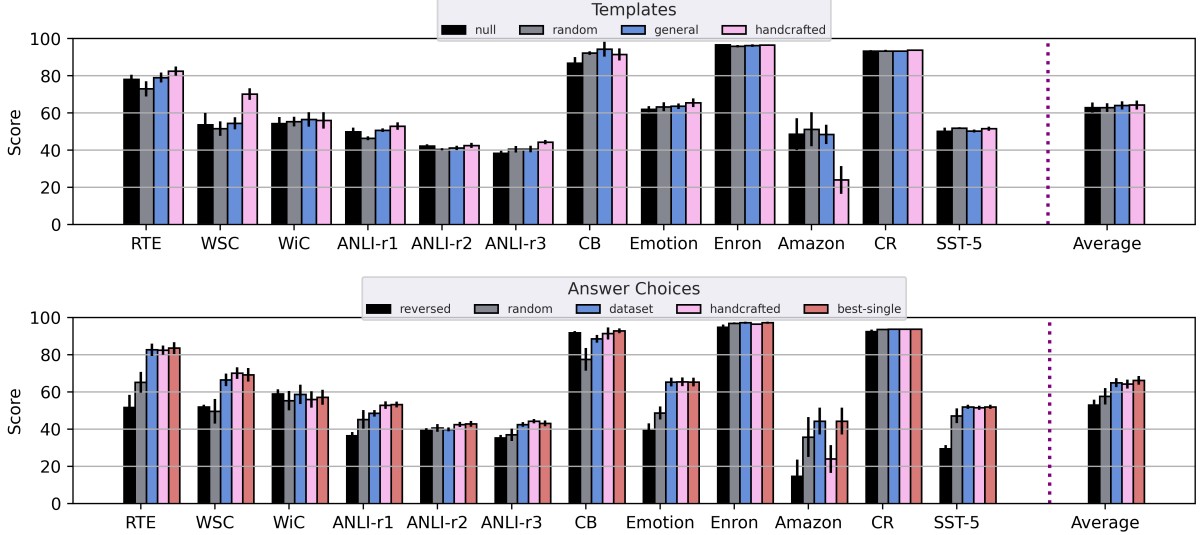

Figure 3: An analysis of prompts used in PEFT of upstream models (here T0), broken down into templates (top) and answer choices (bottom). Experiments span 12 datasets and 8 tasks. Error bars indicate one standard deviation across 5 runs. General task-unspecific templates perform surprisingly well and instruction-independent single answer choice configurations (i.e. dataset and best-single) outperform handcrafted prompts.

masked language models is crucial, Webson and Pavlick (2022) observe that the semantic relevance of a prompt is not a strong performance indicator for upstream models. However, their analysis is restricted to natural language inference whilst using the PET (Schick and Schütze, 2021) algorithm to train the model. Yet, results in Schick and Schütze (2022) suggest that templates do matter in principle, but PET is robust when correctly configured.

These results raise questions regarding the role of prompts for upstream models in the context of automated few-shot learning on unseen tasks. We conduct a systematic ablation study for both templates $\Phi$ and answer choices $\Psi$. We use T-Few with the T0 upstream model and 32 samples per class. We evaluate 12 datasets, spanning 8 tasks. For details on the datasets, see Appendix A.

**Templates.** We design four experiments to understand the importance of accurate task descriptions (i.e. semantics) in increasing order: concatenation of a sample's content without any additional text (null), uniform sampling of words from the training vocabulary (random), general purpose instructions (e.g. *Given ..., the answer is ...*) that are not tailored to the task (general), handcrafted instructions (handcrafted). We use the same handcrafted answer choices and templates across all settings (and vice versa the same templates across experiments for answer choice experiments).

As seen in Figure 3 (top), with a mean score

of 62.8, 62.9, 64.0, 64.2, for each setting, respectively, we observe that **simple task-unspecific templates perform surprisingly well**, only performing slightly worse than more complex handcrafted ones. Templates that are not well-formed or lack an instruction entirely perform substantially worse than handcrafted ones. Note that results differ heavily between datasets. While some datasets (Enron and CR) are virtually unaffected by the design of the template, performance is strongly affected by the template for some other (e.g. RTE, WSC, Amazon).

**Answer Choices.** Similarly, for answer choices we run four experiments: reversed handcrafted answer choices (reversed), uniform sampling of a random word from the training vocabulary (random), label text as presented in a dataset itself, such as *Entailment* in Figure 2 (dataset), and handcrafted choices. Different handcrafted templates for the same task might have different answer choices, depending on the instruction. In contrast, there exists only a single answer choice configuration for dataset answer choices (i.e. mapping from categorical label to text), which we use across all templates.

We observe that **unlike templates, the selection of answer choices makes a large difference in performance**. However, datasets that were particularly robust regarding template design appear to be also robust here. Moreover, despite dataset choices (e.g. *entailment, not_entailment*) not matching a template's instruction (e.g. "*Given ... does ... fol-*

*low? Yes or No?*"), and only having one configuration of choices, we observe comparable performance to handcrafted ones. Thus **neither template-tailored answer choices nor multiple distinct answer choice configurations are needed**. By manually selecting a single configuration of answer choices from both dataset and handcrafted choices (best-single), we easily achieve the highest average score with 66.2. An automated selection mechanism of a single configuration can subsequently perform favourably over multiple distinctly handcrafted prompts.

## 4   AuT-Few: Automated Few-shot Classification with Upstream Models

AuT-Few is a simple, yet efficient, algorithm to automate prompts for upstream models, drawing from the insights gained from Section 3. Figure 2 shows an illustration of AuT-Few's template and answer choice automation. AuT-Few deploys a lightweight template automation approach since accurate task templates are not essential to performance. It selects suitable templates from the collection of prompts the upstream model was instruction-finetuned on (Section 4.1).

On the other hand, the selection of answer choices has a substantial impact on performance. Searching over all possible answer choices is intractable for large upstream models and also imprecise due to the small training size. Thus, AuT-Few only considers two distinct types of answer choices (Section 4.2). One is tailored to the retrieved templates by measuring the log-likelihood on the training data (template-tailored). The other is based on capturing the topic of samples belonging to the same class (topic-specific).

We select the most appropriate template and answer choice configurations via cross-validation. The automated prompts are then used for training and inference of our upstream model, where we largely follow T-Few (c.f. Section 5.1 for details).

### 4.1   Automated Templates via Retrieval

We retrieve templates that are used in instruction tuning the upstream models. This enables us to (i) adhere closely to instructions the model is familiar with and has already learned (ii) exploit the associated inductive bias on answer choices for candidate generation in the next step. Specifically, we consider the collection of all prompts used for instruction tuning, $\Phi_{IT}$, such as the ones shown

in Figure 1 for sentiment classification and paraphrase identification. We then aim to find templates $\Phi_A \subset \Phi_{IT}$ from the collection that are related to our downstream task. For instance, given the NLI sample from Figure 2, we rather want to retrieve templates about paraphrase identification than sentiment classification. The former is both semantically and structurally more similar to NLI, as both have two arguments in their input. For NLI they are *hypothesis* and *premise* while for paraphrase identification these are the two compared sentences.

To find suitable templates, we first filter the collection $\Phi_{IT}$ to templates that match the target task format the most. We achieve this by matching the number of underlying arguments of our target task, against the number of arguments of individual templates in $\Phi_{IT}$. We then do a semantic search via an efficient retrieval system: we query a concatenation of a sample's argument descriptions (e.g. the strings *hypothesis* and *premise*) against all suitable templates in $\Phi_{IT}$ by encoding both query and every template in the collection with a lightweight bi-encoder (Reimers and Gurevych, 2019). If the field descriptions are uninformative (e.g. numbers), we instead use the averaged representations of all samples in $D_{\text{train}}^{tgt_c}$ as the query. Using cosine similarity, we then select the top $R$ templates. Finally, we adjust the retrieved templates to the downstream task via regular expressions to obtain $\Phi_A$.

### 4.2   Automated Selection of Answer Choices

**Generation of Answer Choice Candidates.** Apart from the label descriptions that appear in the dataset, which may not be meaningful, we consider the generation of two distinct types of answer choices given the retrieved templates: template-tailored and topic-specific answer choices. Template-tailored answer choices are generated by finding individual tokens for each class $c$ that maximize the conditional likelihood over the training data of that class $D_{train}^c$, given the retrieved templates $\phi \in \Phi_A$, computed via the upstream model:

$$\mathcal{L}_c = \sum_{x \in D_{train}^c} \sum_{\phi \in \Phi_A} \log p_\theta(v \mid x, \phi),$$

with $v \in \mathcal{V}$ being a token of the subword vocabulary of the upstream model. Tokens unspecific to an individual class might be ranked high across multiple classes. Thus, we further compute for every token how far its likelihood deviates from the mean $\frac{1}{|C|} \sum_{c \in C} \mathcal{L}_c$. We finally select the top-ranked dis-

tinct tokens across all classes that maximize the sum of these scores.

Relying exclusively on the likelihood signal (and the retrieved templates) to find answer choices might amplify the inductive bias of the model and it restricts other potentially viable answer choices [2]. Since our analysis indicates that answer choices not tailored to the templates can still perform strongly, we additionally consider topic-specific answer choices not generated via our upstream model. We use the high quality contextual representations of Sentence Transformers to find single-word (not token) representations that semantically express the underlying content for each class. For each sentence $S_c$ for a particular class, we obtain a contextual representation of the sentence and each word. For every class and over the training vocabulary we then compute the cosine similarity between each sentence and word. We remove words that occur across different classes and finally use the top word for each class as the topic-specific choices.

**Selection of Best Answer Choice Configuration.** We are now tasked to find the best representation for the given task. For each choice option, we consider a joint signal derived from a supervised evaluation, i.e. $F_1$ score, on a subset of the training data $D_{train}$, and from a measure of the overall log probabilities on the test data $D_{test}$. The assumption for the latter is that representative answer choices better estimate the task's distribution, resulting in overall higher log probabilities on unseen data of the target task: $\sum_y \sum_{\phi_A \in \Phi_A} \sum_{x \in D_{test}} (\frac{1}{T} \sum \log p_\theta(\psi_p(y) \mid x, \phi, \psi_p(y)_{<t})$, with $\psi_p$ being the current answer choices configuration. We compute the final score for each candidate by summing the normalized scores of each metric over 3-fold cross-validation.

## 5 Evaluation

### 5.1 Experimental Setup

This section provides an overview of our experimental setup. We are sampling $K$ training samples for each class $y_i \in \mathcal{Y}$, for a total of $K \times |\mathcal{Y}|$ training samples[3]. We do not consider a validation set to exist for hyperparameter-tuning, following Alex

et al. (2021). For baselines, and implementation specifics, including hyperparameters, see Appendix B. For used datasets, see Appendix A.

**Datasets.** We conduct experiments on a total of 12 text classification datasets, spanning a total of 8 tasks. This collection is in essence a combination of evaluation datasets used in Liu et al. (2022) and Tunstall et al. (2022), minus datasets that we consider not traditional classification tasks, e.g. sentence completion, where the meaning of the class changes per instance.

**Implementation Details.** AuT-Few largely follows T-Few (Liu et al., 2022) for finetuning, with some modifications to training and inference to increase robustness for our automated few-shot method. Instead of only learning rescaling vectors of the upstream model's weights ((IA)$^3$), we additionally learn and re-scale decomposition matrices (LoRA), as proposed by Hu et al. (2022a). (IA)$^3$ and LoRA are complementary and the gradient updates from both methods can be made persistent to the model's weights after training without inquiring additional inference costs over the upstream model itself. Another limitation of T-Few is its inference algorithm. T-Few selects a single template at random (c.f. Section 2) and it can be a poor approximation of the overall expectation, especially with noisy templates as used with AuT-Few. We instead run a Monte-Carlo approximation over all retrieved templates, computing a weighted average over the probabilities computed via each template.

**Baselines.** In addition to the current state-of-the-art few-shot learning method T-Few, we consider SetFit (Tunstall et al., 2022) (with a RoBERTA backbone), which is of particular relevance in our context, since it is the state-of-the-art efficient prompt-free few-shot method. We also compare against a fully-finetuned RoBERTa$_{LARGE}$ model, based on the baseline in Tunstall et al. (2022). The majority baseline is based on the class distribution in the test data.

### 5.2 Results

Results for $K = 32$ samples per class are shown in Table 1. Both T-Few and AuT-Few use T0-3B as the upstream model. We report accuracy on all datasets with the exception of Amazon-CF, where we report Matthew's correlation coefficient due to the skewed distribution, following Tunstall et al. (2022).

---

[2]For example the input prompt and samples might have been encountered for NLI tasks, focusing on options working particularly well for this scenario.

[3]While in Liu et al. (2022) samples are drawn randomly, i.e. not stratified, we largely adhere to the traditional N-Way-K-shot classification setting, as data imbalance in training is an aspect to be explored separately.

| | **Majority** | **Zero-shot** | **Finetune** | **SetFit** | **Rand. T-Few** | **T-Few** | **AuT-Few** |
|---|---|---|---|---|---|---|---|
| RTE | 52.7 | $65.6_{1.2}$ | $56.4_{5.6}$ | $51.4_{1.8}$ | $65.2_{5.6}$ | $\mathbf{82.5}_{2.4}$ | $81.4_{2.4}$ |
| WSC | 63.5 | $62.1_{3.9}$ | $49.2_{7.1}$ | $50.3_{4.4}$ | $49.6_{6.6}$ | $\mathbf{70.2}_{3.1}$ | $59.2_{1.5}$ |
| WiC | 50.0 | $51.3_{0.6}$ | $53.9_{5.1}$ | $55.0_{5.1}$ | $55.3_{5.2}$ | $\underline{55.9}_{4.4}$ | $\mathbf{58.4}_{5.1}{}^{*}$ |
| ANLI-R1 | 33.4 | $35.6_{0.8}$ | $32.1_{1.9}$ | $32.9_{1.6}$ | $45.2_{4.9}$ | $\mathbf{52.9}_{2.0}$ | $\underline{49.1}_{3.7}{}^{*}$ |
| ANLI-R2 | 33.4 | $33.6_{0.7}$ | $33.4_{1.6}$ | $34.0_{1.7}$ | $40.6_{2.0}$ | $\mathbf{42.5}_{1.4}$ | $\underline{42.0}_{1.5}$ |
| ANLI-R3 | 33.5 | $34.2_{0.8}$ | $31.5_{1.6}$ | $32.7_{1.0}$ | $36.9_{3.4}$ | $\mathbf{44.2}_{1.2}$ | $\underline{43.5}_{3.0}$ |
| CB | 50.0 | $57.5_{0.8}$ | $86.1_{6.6}$ | $84.3_{5.0}$ | $77.5_{6.1}$ | $\underline{91.4}_{3.2}$ | $\mathbf{93.9}_{1.6}$ |
| Emotion | 35.2 | $42.1_{0.8}$ | $57.6_{\,3.5}$ | $\underline{71.9}_{3.2}$ | $48.7_{3.5}$ | $65.4_{2.3}$ | $\mathbf{72.6}_{2.5}{}^{*}$ |
| Enron | 50.9 | $53.3_{0.4}$ | $92.2_{2.4}$ | $95.1_{1.2}$ | $\mathbf{96.9}_{0.6}$ | $\underline{96.5}_{0.4}$ | $95.5_{0.5}$ |
| Amazon-CF | 0.00 | $0.04_{0.7}$ | $40.5_{9.9}$ | $\mathbf{60.1}_{3.0}$ | $35.7_{10.6}$ | $24.0_{7.5}$ | $\underline{59.0}_{8.2}{}^{*}$ |
| CR | 64.2 | $88.9_{0.4}$ | $84.8_{4.3}$ | $90.7_{1.7}$ | $\underline{93.6}_{3.5}$ | $\mathbf{93.7}_{0.2}$ | $92.5_{1.1}$ |
| SST-5 | 26.3 | $38.9_{1.0}$ | $42.1_{3.4}$ | $\underline{49.2}_{0.9}$ | $47.2_{3.9}$ | $\mathbf{51.5}_{1.1}$ | $48.6_{2.5}$ |
| Average ↑ | 41.1 | $47.3_{1.0}$ | $55.0_{4.4}$ | $59.0_{2.6}$ | $57.7_{4.4}$ | $\underline{64.2}_{2.4}$ | $\mathbf{66.3}_{2.5}$ |

Table 1: Main results with 32 samples per class, averaged over five runs. AuT-Few adopts T0 as the upstream model. *Rand. T-Few* uses randomly selected answer choices. Statistically significant differences between AuT-Few and T-Few are marked with *, using a two-sided Monte-Carlo permutation test with 10000 repetitions (p < 0.01). AuT-Few has the highest average score across datasets without the use of handcrafted task prompts while maintaining comparable standard deviation to T-Few and SetFit.

AuT-Few outperforms T-Few ($64.2 \pm 2.4$) and SetFit ($59.0 \pm 2.6$), with an average score of $66.3 \pm 2.5$. A trivial T-Few automation strategy that randomly draws answer choices from the training data (c.f Section 3) performs substantially worse than AuT-Few with much higher variability ($57.7 \pm 4.4$). While AuT-Few has a higher average score than T-Few, the latter wins against AuT-Few on 8 out of 12 datasets. However, we observe a statistically significant difference[4] on only 4 datasets. Out of these four datasets where we observe statistical significance, AuT-Few outperforms T-Few in three of them (WiC, Emotion, Amazon-CF).[5] Moreover, we would like to emphasise that performing even comparable against T-Few is already a win since the latter uses multiple diverse handcrafted prompts for each target task while AuT-Few does not require any manual involvement by the user to optimize the prompt while maintaining comparable standard deviation.

On the blind test set with the best variant of T0 (T0++, 11B parameters) AuT-Few achieves an average score of 71.3 versus 70.5 for T-Few (with the same backbone), excluding WiC and WSC, as these datasets have been used to train T0 (see App. C.1 for detailed scores).

We note that the automated prompts are not always semantically coherent. As shown in Appendix D, automation choices for some datasets, such as *mp3player* and *ipod* for CR, appear odd, yet the model still achieves a very high score on them. This observation can be explained by our findings in section 3, identifying that some datasets such as CR and EnronSpam are particularly robust towards the task description *and* the answer choices. For CR, AuT-Few's cross-validation strategy for selecting the best answer choice subsequently measures almost identical scores for all three choice configurations (90.1, 89.8, 90.4 for the dataset, template-tailored, and topic-specific choices, respectively), resulting in the seemingly erroneously answer-choice selection.

**Results across Upstream Models & Efficiency.** Results of AuT-Few with different upstream models, namely BART0, T0, and Flan-T5 are seen in Table 2. The results in the table are computed without Monte-Carlo approximation, resulting in a minor performance decline, yet simplifying the efficiency comparison. Datasets that are part of the instruction-tuning corpus of Flan-T5 or BART0 have been excluded (greyed out). BART0 being about 8 times smaller than T0 performs substantially worse, but it still substantially outperforms T-Few with BART0 and maintains a higher average score than SetFit. Flan-T5 performs on average the best on its unseen datasets, indicating the improved

---

[4]We ran the two-sided Monte-Carlo permutation test with 10000 repetitions (p-value < 0.01). Significance for a dataset holds iff results are significant across *all* seeds.

[5]Notably, the performance difference between AuT-Few and T-Few on WSC, the only dataset where AuT-Few performs substantially worse, is not statistically significant given our test: this can be explained by the very small sample size of the dataset's evaluation data of only 104 samples. Liu et al. (2022) also observed "unstable results" on WSC, see the discussion on this Github issue.

| | | T-Few | | AuT-Few | | |
|---|---|---|---|---|---|---|
| | **SetFit** | **BART0** | **T0** | **BART0** | **T0** | **Flan-T5** |
| # Param. | 330M | 400M | 3B | 400M | 3B | 3B |
| # Tr. Param. | 330M | 0.1M | 0.3M | 1.9M | 10.5M | 10.5M |
| Inf. FLOPs | 2.5e10 | 1.9e10 | 1.8e11 | 1.9e10 | 1.8e11 | 1.8e11 |
| Tr. FLOPs | 8.5e14 | 4.1e14 | 3.9e15 | 3.6e15 | 2.7e16 | 2.7e16 |
| RTE | $51.4_{1.8}$ | $80.4_{1.5}$ | $82.5_{2.4}$ | $71.3_{7.0}$ | $79.3_{3.5}$ | $90.1_{1.8}$ |
| WSC | $50.3_{4.4}$ | $61.2_{3.3}$ | $70.2_{3.1}$ | $52.9_{3.1}$ | $58.3_{4.6}$ | $73.1_{5.6}$ |
| WiC | $55.0_{5.1}$ | $59.4_{1.5}$ | $55.9_{4.4}$ | $55.1_{2.9}$ | $59.7_{4.9}*$ | $67.6_{2.5}$ |
| ANLI-R1 | $32.9_{1.6}$ | $34.7_{0.7}$ | $52.9_{2.0}$ | $33.4_{3.1}$ | $47.8_{3.5}*$ | $67.1_{3.5}$ |
| ANLI-R2 | $34.0_{1.7}$ | $34.7_{1.0}$ | $42.5_{1.4}$ | $36.1_{1.7}$ | $42.1_{1.1}$ | $53.3_{2.6}$ |
| ANLI-R3 | $32.7_{1.6}$ | $36.9_{1.3}$ | $44.2_{1.2}$ | $36.2_{1.1}$ | $42.1_{2.9}$ | $52.1_{2.8}$ |
| CB | $81.3_{5.0}$ | $78.6_{7.3}$ | $91.4_{3.2}$ | $85.7_{4.6}$ | $93.6_{1.6}$ | $91.0_{1.3}$ |
| Emotion | $71.9_{3.2}$ | $42.0_{3.3}$ | $65.4_{2.3}$ | $63.9_{6.5}$ | $72.1_{2.6}*$ | $74.3_{1.8}$ |
| Enron | $95.1_{1.2}$ | $54.3_{1.6}$ | $96.5_{0.4}$ | $92.8_{1.8}$ | $95.6_{1.8}$ | $96.1_{0.7}$ |
| Amazon-CF | $60.1_{3.0}$ | $0.02_{3.0}$ | $24.0_{7.5}$ | $55.0_{11.0}$ | $59.4_{6.8}*$ | $62.7_{7.5}$ |
| CR | $90.7_{1.7}$ | $91.7_{0.8}$ | $93.7_{0.2}$ | $90.6_{0.8}$ | $92.0_{1.5}$ | $93.2_{0.3}$ |
| SST-5 | $49.2_{0.9}$ | $42.4_{0.3}$ | $51.5_{1.1}$ | $47.4_{3.9}$ | $47.7_{1.3}$ | $48.6_{7.2}$ |
| Average ↑ | $59.9_{2.2}$ | $49.8_{2.0}$ | $64.5_{2.2}$ | $61.2_{4.1}$ | $67.1_{3.0}$ | – |

Table 2: Results and computational costs using different upstream models, 32 samples per class. All results are computed without Monte-Carlo approx. Datasets that appear in an upstream model's training are greyed out. WiC and WSC were excluded from *all* averages.

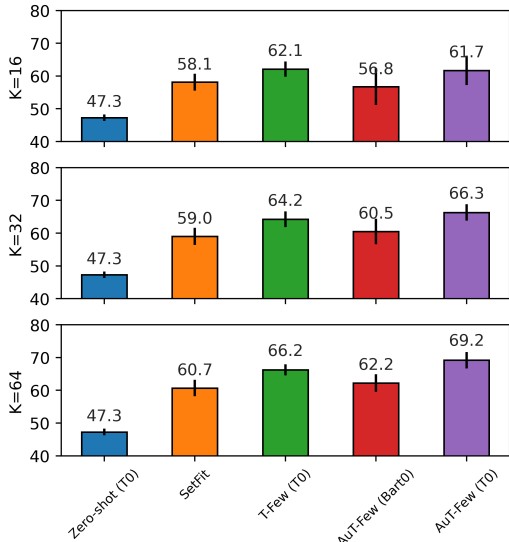

Figure 4: Average scores when finetuned on 16, 32, and 64 samples per class. AuT-Few performs better relative to the baselines with more training samples.

capabilities of the model's much larger and diverse instruction-tuning. These results highlight the effectiveness of AuT-Few across upstream models of varying sizes.

The computational costs for training and inference are listed in Table 2. We follow the approach adopted by Liu et al. (2022) and Tunstall et al. (2022) to measure computational costs, namely FLOPs-per-token (Kaplan et al., 2020). AuT-Few requires about 7x the training cost of T-Few, yet remains computationally accessible, taking only a few hours to train on a single A10G GPU, since the number of training steps for few-shot PEFT is overall small. Similarly, while AuT-Few with BART0 takes 4.2x longer than SetFit, it still takes less than an hour of total training time. Importantly, during inference, AuT-Few is as efficient as T-Few (excluding Monte-Carlo approximation, otherwise scaling linearly with the number of retrieved templates). AuT-Few with BART0 is even more efficient than SetFit during inference, requiring only 60% of its computation while maintaining a competitive score.

We emphasize that while T-Few takes somewhat less computation than AuT-Few, T-Few requires significantly more human intervention, and human time is much more valuable than computer time. The difference of a couple hours of computer time is negligible when it can save orders of magnitude more human time and associated costs.

**Varying sample sizes.** Figure 4 shows the performance of our baselines as well as Aut-Few over 16,

32, and 64 samples, respectively. With $K = 16$, we observe slightly worse performance than T-Few with AuT-Few. The provided signal from only 16 samples is too noisy for our automation pipeline. With an increase in number of training samples follows a larger lead of AuT-Few over other models. While AuT-Few (T0) is on average 2.1 points better than T-Few with 32 samples, this lead increases to 3.1 for $K = 64$. Similar observation is made when comparing AuT-Few (BART0) with SetFit.

**Real-world evaluation: RAFT.** RAFT (Alex et al., 2021) is a benchmark targeted towards evaluating few-shot classification methods. It consists of 11 datasets, from various domains, such as the legal or medical domain. In RAFT 50 randomly sampled training samples are provided, with a potentially imbalanced label distribution. We submitted predictions of AuT-Few with the 11B Flan-T5 backbone, with handcrafted prompts as provided by RAFT (AuT-Few (H)), as well as with our automated prompts (AuT-Few). We do not make any manual dataset adjustments, with the exception of Banking_77 as only a subset of the classes appears in its training data, c.f. App. C.2.

Results are shown in Table 3. Our method with handcrafted prompts and the Flan-T5 upstream model achieves rank-1 with the overall highest average score. Our automated version achieves scores slightly below T-Few (the previously 2nd ranked system). This is largely due to AuT-Few's poor performance on a single dataset, Tweet-Eval-

Hate, as a result of improper selection of answer choices. However, AuT-Few has the best average rank across all five models with 2.45. It wins against T-Few on 7 out of 11 datasets. Furthermore, it has the highest overall win rate, winning against all other models we considered (including our approach with handcrafted prompts) on 4 out of 11 datasets, see Table 7. These results highlight AuT-Few's robustness and generalizability to real-world classification tasks.

| Rank | Method | Avg. Score ↑ | Avg. Rank ↓ |
|---|---|---|---|
| – | AuT-Few (H) | **77.3** | 2.82 |
| – | AuT-Few | 74.7 | **2.45** |
| 1 | yiwise | 76.8 | 2.55 |
| 2 | T-Few | 75.8 | 2.82 |
| 12 | SetFit | 71.3 | 4.27 |
| 5 | Human baseline | 73.5 | – |

Table 3: Results on the RAFT benchmark as of October 19 2023. Avg. Rank is reported across the shown models. Our method with handcrafted prompts achieves rank-1 with the overall highest average score while AuT-Few has the best average rank and highest win rate.

**Ablation.** Results of our ablation study for AuT-Few with 32 samples per class are shown in Table 4. We ablate our template retrieval method by considering randomly selected templates from the instruction tuning KB, as well as template retrieval from the entire PromptSource collection of prompts. As seen both settings perform worse than AuT-Few, with higher standard deviation across seeds. While retrieving from the entire collection performs slightly better for tasks that appear in it (e.g. NLI, emotion classification), it strongly underperforms on unseen ones (e.g. WiC, Amazon-CF). Further, the ablation of the choice options shows that each definition of answer choices by itself performs worse than AuT-Few (including the label descriptions that appear in the dataset). Finally, we see that our modifications to T-Few's inference and training are effective, with both LoRA and $(IA)^3$ PEFT performing worse individually. Note that AuT-Few still outperforms T-Few even when using only $(IA)^3$, indicating AuT-Few's superiority without any architectural adjustments.

## 6 Conclusion

AuT-Few replaces hand-designed task-specific prompts with automated templates, and achieves state-of-the-art results on a wide range of datasets

| Setup | | Avg. Score |
|---|---|---|
| AuT-Few | | $66.3_{2.5}$ |
| Template | w/o retrieved template (randomized) | $65.7_{2.9}$ |
| | w/ entire Collection | $65.6_{2.9}$ |
| Choices | only dataset | $65.5_{2.7}$ |
| | only template-tailored | $63.3_{3.4}$ |
| | only topic-specific | $62.2_{4.3}$ |
| Improv. | w/o Monte-Carlo approximation | $65.8_{3.0}$ |
| | only LoRA | $63.4_{3.4}$ |
| | only $(IA)^3$ | $65.2_{2.5}$ |

Table 4: Ablation for AuT-Few with 32 samples per class: *randomized* indicates randomly selected templates, *entire Coll.* considers all PromptSource prompts.

and tasks, and the best average rank across datasets on the RAFT benchmark. Machine learning, especially few-shot learning, is about automation. Although T-Few takes less computation, it requires hand-designed prompts which involves significant human intervention and expertise. Human-time is profoundly more valuable than computer time, and AuT-Few saves this valuable human time while still retaining computational tractability. Future work includes the identification of causes for the observations made in section 3, particularly for datasets that are completely unaffected by the prompt's design (e.g Enronspam and CR).

## Limitations

This work and the automation pipeline is constrained to classification tasks in English. The role of templates and answer choices is necessarily different for tasks such as natural language generation (e.g. summarization or question answering) where a single textual class representation does not exist. The proposed automated few-shot approach is not expected to work well under extremely low data regime or when training samples are highly imbalanced (i.e. < 8 samples per class) as some data signal is required for optimizing the choice space. While our evaluation aims to cover a diverse range of classification tasks, the list of evaluation tasks is not exhaustive. Subsequently, there is no guarantee that AuT-Few performs equally well on every unseen tasks, particular ones that divert strongly from tasks the model has seen during instruction tuning.

## Ethics Statement

Our paper makes state-of-the-art few-shot classification methods more accessible to non-experts for real-world problems. The goal of this work is not to replace human involvement in the deployment of AI systems but instead to shift human resources to

other essential aspects of model deployment such as the analysis of data, biases, or system errors. We discussed the computational costs of our automation approach and show that they are comparable at similar model size with the most efficient few-shot systems, which themselves again are computationally much more efficient than full-data and full model fine-tuning, or in-context learning.

## Acknowledgements

The authors would like to thank Lewis Tunstall for his help to submit AuT-Few's predictions to the RAFT leaderboard and Aditya Rawal for pointing us to relevant related work. We would also like to thank the anonymous reviewers for their time and effort giving us valuable feedback on our paper.

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

# A   Datasets

We conduct experiments on a total of 12 text classification datasets. The tasks we consider are 1) natural language inference: RTE (Dagan et al., 2005) , CB (de Marneffe et al., 2019), ANLI (Nie et al., 2020); 2) coreference resolution: WSC (Levesque et al., 2012); 3) word sense disambiguation: WiC (Pilehvar and Camacho-Collados, 2019); 4) counterfactual detection: Amazon-CF (O'Neill et al., 2021); 5) sentiment classification: SST-5 (Socher et al., 2013), Customer reviews (CR) (Conneau and Kiela, 2018); 6) emotion classification: emotion (Saravia et al., 2018); and 7) spam detection: Enron (Metsis et al., 2006). All datasets are in English. Enron contains personal identifiable information, yet substantial efforts have been made to remove any integrity problems and samples of affected employees, see here for reference. Enron is an long-established dataset to use for classification problems and our use is in line with previous usages of it.

# B   Implementation Details

**Parameter-efficient fine-tuning via low-rank adaptation and rescaling** While exclusively rescaling weights of an upstream model via IA$^3$ has shown to perform remarkably well, the expressiveness of the fine-tuning process is restricted, due to $\Delta h$ (the accumulated gradient update) being always of the form $\mid W_0 - \lambda W_0 \mid$, with $W_0$ being the weights of the upstream model and $\lambda$ being the rescaling vector. For tasks that require major adaptation capabilities, this might pose a hindrance. In contrast, LoRA explicitly models via decomposition matrices the gradient update $\Delta h = BA$, resulting in higher expressiveness (about 10x as many parameters as IA$^3$), but has repeatedly shown in our experiments to have substantially higher variability. We hence combine both PEFT strategies, by rescaling both the weights of the upstream model and the accumulated gradient updates jointly: $h = \lambda(W_0 x + BAx)$. After training, both $\lambda$ and $BA$ can be applied to $W_0$, making the weight updates persistent without inquiring any additional computation during inference. Following Liu et al. (2022), we pre-train the weights of the rescaling vectors in a similar fashion to the upstream model. While the authors only train the vectors for 100K steps, we observed further improvements when training them for longer (500K steps).

**Inference via Monte-Carlo Approximation over Templates** As outlined in section 3, in its current version the expectation over template and choice space is approximated during inference by randomly drawing a template from a collection of handcrafted ones. Besides being non-deterministic, the selected template might be a poor approximation of the overall expectation. Instead, we run a Monte-Carlo Approximation over the template space $\Phi_A$, by computing a weighted average over all retrieved templates:

$$\hat{y} = \text{argmax}_y E_{\Phi,\Psi}[p_\theta(y_i \mid x, \Phi, \Psi)]$$
$$= \text{argmax}_y \sum_{r=1}^{R} w_r p_\theta(y_i \mid x, \phi_r, \psi_A),$$

with $\sum_{r=1}^{R} w_r = 1$. We determine the weights for each template by computing the log-likelihood of each template on $D_{test}$ and applying a softmax function on them, following the previously mentioned motivation.

**Hyperparameters** Since our joint PEFT method converges substantially faster than IA$^3$ by itself, we set the number of total steps to 600 (contrary to 1000 used by T-Few). Further, for both T-Few and AuT-Few we use the following hyperparameters across all experiments: we use Adam, a learning rate of $1^{-3}$, cosine decay with a warmup ratio of 0.06, a learning rate decay of 1.0, and a batch size of 8. The contextual embeddings for template retrieval as well the topic-specific choices are generated using sentence-transformers' `all-MiniLM-L6-v2` encoder model. For all main experiments, we set the number of retrieved templates to $R = 5$. The underlying prompt knowledge base used is PromptSource (Bach et al., 2022). For selecting the best answer choices, we split the training data using 3-fold cross-validation and train the upstream model with identical hyperparameters as our final model for every choice option.

**System & Code** All models (550M, 3B, 11B parameters) are trained and run on a single A10G GPU with 23GB of memory by using gradient checkpointing, bfloat16 floating-point format, and in the case of the 11B model by offloading parameters using DeepSpeed [6]. We produce results for the SetFit and finetune baseline using the associated repository[7]. We filter stopwords and punctuation from the vocabulary of topic-specific answer

choices using NLTK (Bird and Loper, 2004). Our code and models will be made openly accessible under Apache License 2.0.

**Baselines** In addition to the current state-of-the-art (Liu et al., 2022), we consider SetFit (Tunstall et al., 2022), as well as a standard finetuned LM. SetFit is of particular relevance to us since it is the state-of-the-art prompt-free few-shot method, shown to perform competitively to T-Few in their experiments while being computationally substantially more efficient. In their comparison to T-Few, they use a very small variation of the sentence-transformer MPNET, consisting of only 110M, however, we observed substantially better performance with the larger ROBERTa sentence-transformer model (355M parameters). Hence, we report results on the latter model[8]. The traditionally finetuned model is a RoBERTa$_{\text{LARGE}}$ model, fully-finetuned with an additional linear head, based on the baseline in (Tunstall et al., 2022).

## C Detailed Results

Detailed results of the experiment in Table 1 for different sample sizes are shown in Table 8.

### C.1 Results Blind Test Set

| Dataset | T-Few | AuT-Few |
|---------|-------|---------|
| RTE | 87.2 | 82.1 |
| ANLI-R1 | 60.7 | 54.6 |
| ANLI-R2 | 52.1 | 49.1 |
| ANLI-R3 | 51.9 | 51.8 |
| CB | 93.6 | 96.0 |
| Emotion | 62.1 | 71.7 |
| Enron | 97.0 | 97.6 |
| Amazon-CF | 50.2 | 62.6 |
| CR | 93.7 | 92.8 |
| SST-5 | 56.6 | 55.1 |
| Avg | 70.5 | 71.3 |

Table 5: Results on the held out test set. Excluding WiC and WSC as these wereeen in T0pp's pre-training.

---

[6]https://github.com/microsoft/DeepSpeed
[7]https://github.com/huggingface/setfit

---

[8]Albeit some sentence-transformer models are targeted for a certain domain, e.g. QA or NLI, in our experimental setup we aim to minimize human involvement, including model selection. Hence, the same pre-trained model is used across all experiments.

## C.2 Results RAFT

RAFT consists of 11 datasets from different domains. The individual datasets included in RAFT are ADE Corpus V2 (medical case reports) (Gurulingappa et al., 2012), Banking77 (Casanueva et al., 2020), NeurIPS impact statement risks (Ashurst et al., 2022), Onestop English (Vajjala and Lučić, 2018), Overruling (legal domain) (Zheng et al., 2021), Systematic Review Inclusion (Saeri et al., 2022), Tai safety research (Riedel and Deibel, 2020), Terms of Service (Lippi et al., 2019), Tweet Eval Hate (Basile et al., 2019), and Twitter Complaints (Preotiuc-Pietro et al., 2019). All datasets are in English.

Since only a small subset of the 77 classes appear in the training data of the Banking_77 dataset, we directly use the dataset's class representations for the answer choices. Banking_77 is strictly speaking a zero-shot and few-shot evaluation dataset and previous work such as SetFit that does not use a verbalizer at all also had to make use of the given class representations for that dataset[9].

## D   Automated Choices

The generated and selected answer choices as used in AuT-Few with $K = 32$ and T0 as the upstream model on seed 0 are shown in Table 9.

## E   Automated Templates

The retrieved templates as used in AuT-Few with $K = 32$ and T0 as the upstream model on seed 0 are shown in Table 10.

---

[9]https://towardsdatascience.com/
sentence-transformer-fine-tuning-setfit-\
outperforms-gpt-3-on-few-shot-text-class\
ification-while-d9a3788f0b4e

| Dataset | Handcrafted Dataset Choice | Automated Choice |
|---|---|---|
| Ade | ADE-related/not ADE-related | chemotherapyinduced/diagnosis |
| Banking | c.f. C.2 | c.f. C.2 |
| Neurips | doesn't mention a harmful application/mentions a harmful application | doesn't mention a harmful application/mentions a harmful application |
| One Stop | elementary/intermediate/advanced | Black/World/Science |
| Overruling | not overruling/overruling | court/overrule |
| Org Types | company/research institute/university | company/research institute/university |
| Review | included/not included | included/not included |
| Tai Safety | TAI safety research / not TAI safety research | agent/learning |
| ToS | not potentially unfair/potentially unfair | not potentially unfair/potentially unfair |
| Eval Hate | hate speech/not hate speech | Sports/World |
| Complaints | complaint/no complaint | complaint/no complaint |

Table 6: Generated answer choices, when using T0 and 32 samples for seed 0.

| System | Ade | Banking | Neurips | One Stop | Overruling | Org Types | Review | Tai Safety | ToS | Eval Hate | Complaints |
|---|---|---|---|---|---|---|---|---|---|---|---|
| *AuT-Few (H)* | 0.837 | 0.647 | 0.78 | **0.847** | 0.942 | **0.917** | **0.687** | 0.703 | 0.728 | 0.517 | 0.892 |
| *AuT-Few* | 0.846 | 0.587 | **0.898** | 0.77 | **0.963** | 0.801 | 0.62 | **0.742** | 0.738 | 0.350 | **0.901** |
| *yiwise* | **0.856** | **0.695** | 0.839 | 0.698 | 0.944 | 0.906 | 0.493 | 0.737 | 0.749 | **0.647** | 0.883 |
| *T-Few* | 0.804 | **0.695** | 0.833 | 0.676 | 0.95 | 0.915 | 0.508 | 0.736 | **0.75** | 0.586 | 0.879 |
| *SetFit* | 0.799 | 0.632 | 0.859 | 0.76 | 0.93 | 0.769 | 0.503 | 0.664 | 0.604 | 0.487 | 0.831 |

Table 7: Results on RAFT.

| | Majority | Zero-shot | Finetune | SetFit | T-Few | AuT-Few (H) | AuT-Few (w/o D) | AuT-Few (A) |
|---|---|---|---|---|---|---|---|---|
| RTE | 52.7 | $65.6_{1.2}$ | $50.3_{2.6}$ | $52.7_{4.0}$ | $81.0_{1.5}$ | $81.8_{3.9}$ | $81.0_{2.4}$ | $80.1_{1.5}$ |
| WSC | 63.5 | $62.1_{3.9}$ | $53.7_{4.8}$ | $50.2_{5.3}$ | $61.9_{4.4}$ | $65.0_{5.7}$ | $50.5_{4.8}$ | $48.9_{6.2}$ |
| WiC | 50.0 | $51.3_{0.6}$ | $53.3_{4.1}$ | $57.0_{3.9}$ | $54.4_{2.9}$ | $60.6_{2.5}$ | $52.7_{4.5}$ | $54.9_{4.7}$ |
| ANLI-R1 | 33.4 | $35.6_{0.8}$ | $32.9_{1.4}$ | $32.3_{1.3}$ | $50.2_{2.0}$ | $51.1_{2.4}$ | $47.4_{4.5}$ | $48.0_{3.7}$ |
| ANLI-R2 | 33.4 | $33.6_{0.7}$ | $34.3_{1.0}$ | $34.0_{1.7}$ | $42.4_{0.7}$ | $40.8_{1.9}$ | $41.3_{0.6}$ | $41.1_{1.9}$ |
| ANLI-R3 | 33.5 | $34.2_{0.8}$ | $33.2_{1.9}$ | $32.3_{0.9}$ | $43.0_{1.5}$ | $42.8_{1.8}$ | $36.9_{2.2}$ | $38.1_{5.0}$ |
| CB | 50.0 | $57.5_{0.8}$ | $64.3_{5.2}$ | $81.4_{4.7}$ | $85.7_{2.8}$ | $91.8_{2.0}$ | $85.7_{8.1}$ | $87.5_{7.7}$ |
| Emotion | 35.2 | $42.1_{0.8}$ | $37.5_{4.3}$ | $68.4_{1.7}$ | $62.0_{3.2}$ | $72.6_{3.3}$ | $61.5_{4.1}$ | $66.0_{1.7}$ |
| Enron | 50.9 | $53.3_{0.4}$ | $90.7_{3.5}$ | $94.5_{1.8}$ | $95.6_{0.9}$ | $96.1_{1.4}$ | $93.6_{2.6}$ | $92.7_{3.5}$ |
| Amazon-CF | 0.00 | $0.04_{0.7}$ | $20.4_{12.6}$ | $56.7_{4.0}$ | $23.4_{5.3}$ | $61.7_{12.1}$ | $32.4_{8.5}$ | $38.6_{15.2}$ |
| CR | 64.2 | $88.9_{0.4}$ | $74.7_{8.1}$ | $91.0_{1.1}$ | $93.4_{2.7}$ | $93.6_{0.4}$ | $92.7_{1.3}$ | $92.6_{1.7}$ |
| SST-5 | 26.3 | $38.9_{1.0}$ | $37.5_{5.1}$ | $47.9_{1.4}$ | $51.7_{2.3}$ | $52.1_{0.9}$ | $51.6_{1.4}$ | $51.2_{1.3}$ |
| Average | 41.1 | $47.3_{1.0}$ | $48.5_{4.6}$ | $58.1_{2.6}$ | $62.1_{2.3}$ | $64.4_{3.3}$ | $60.7_{3.7}$ | $61.7_{4.5}$ |

16 samples per class.

| | Majority | Zero-shot | Finetune | SetFit | T-Few | AuT-Few (H) | AuT-Few (A w/o D) | AuT-Few (A) |
|---|---|---|---|---|---|---|---|---|
| RTE | 52.7 | $65.6_{1.2}$ | $56.4_{5.6}$ | $51.4_{1.8}$ | $82.5_{2.4}$ | $81.8_{3.9}$ | $82.3_{4.0}$ | $81.4_{2.4}$ |
| WSC | 63.5 | $62.1_{3.9}$ | $49.2_{7.1}$ | $50.3_{4.4}$ | $70.2_{3.1}$ | $65.0_{5.7}$ | $50.8_{4.4}$ | $59.2_{1.5}$ |
| WiC | 50.0 | $51.3_{0.6}$ | $53.9_{5.1}$ | $55.0_{5.1}$ | $55.9_{4.4}$ | $60.6_{2.5}$ | $55.6_{3.9}$ | $58.4_{5.1}$ |
| ANLI-R1 | 33.4 | $35.6_{0.8}$ | $32.1_{1.9}$ | $32.9_{1.6}$ | $52.9_{2.0}$ | $51.1_{2.4}$ | $50.1_{3.8}$ | $49.1_{3.7}$ |
| ANLI-R2 | 33.4 | $33.6_{0.7}$ | $33.4_{1.6}$ | $34.0_{1.7}$ | $42.5_{1.4}$ | $40.8_{1.9}$ | $42.7_{1.8}$ | $42.0_{1.5}$ |
| ANLI-R3 | 33.5 | $34.2_{0.8}$ | $31.5_{1.6}$ | $32.7_{1.0}$ | $44.2_{1.2}$ | $42.8_{1.8}$ | $42.9_{3.8}$ | $43.5_{3.0}$ |
| CB | 50.0 | $57.5_{0.8}$ | $86.1_{6.6}$ | $84.3_{5.0}$ | $91.4_{3.2}$ | $91.8_{2.0}$ | $93.9_{2.0}$ | $93.9_{1.6}$ |
| Emotion | 35.2 | $42.1_{0.8}$ | $57.6_{3.5}$ | $71.9_{3.2}$ | $65.4_{2.3}$ | $72.6_{3.3}$ | $70.5_{2.2}$ | $72.6_{2.5}$ |
| Enron | 50.9 | $53.3_{0.4}$ | $92.2_{2.4}$ | $95.1_{1.2}$ | $96.5_{0.4}$ | $96.1_{1.4}$ | $95.5_{1.2}$ | $95.5_{0.5}$ |
| Amazon-CF | 0.00 | $0.04_{0.7}$ | $40.5_{9.9}$ | $60.1_{3.0}$ | $24.0_{7.5}$ | $61.7_{12.1}$ | $53.2_{8.3}$ | $59.0_{8.2}$ |
| CR | 64.2 | $88.9_{0.4}$ | $84.8_{4.3}$ | $90.7_{1.7}$ | $93.7_{0.2}$ | $93.6_{0.4}$ | $93.0_{1.3}$ | $92.5_{1.1}$ |
| SST-5 | 26.3 | $38.9_{1.0}$ | $42.1_{3.4}$ | $49.2_{0.9}$ | $51.5_{1.1}$ | $52.1_{0.9}$ | $50.0_{3.2}$ | $48.6_{2.5}$ |
| Average | 41.1 | $47.3_{1.0}$ | $55.0_{4.4}$ | $59.0_{2.6}$ | $64.2_{2.4}$ | $67.5_{3.3}$ | $65.1_{2.9}$ | $66.3_{2.5}$ |

32 samples per class.

| | Majority | Zero-shot | Finetune | SetFit | T-Few | AuT-Few (H) | AuT-Few (A w/o D) | AuT-Few (A) |
|---|---|---|---|---|---|---|---|---|
| RTE | 52.7 | $65.6_{1.2}$ | $52.1_{5.1}$ | $52.3_{3.1}$ | $86.1_{0.4}$ | $85.4_{1.2}$ | $85.2_{2.8}$ | $85.7_{1.9}$ |
| WSC | 63.5 | $62.1_{3.9}$ | $48.8_{2.4}$ | $48.9_{5.3}$ | $71.7_{2.5}$ | $72.7_{4.4}$ | $58.2_{6.1}$ | $65.1_{5.9}$ |
| WiC | 50.0 | $51.3_{0.6}$ | $56.3_{3.5}$ | $56.7_{2.3}$ | $58.2_{3.1}$ | $60.7_{3.3}$ | $56.8_{1.4}$ | $58.7_{3.1}$ |
| ANLI-R1 | 33.4 | $35.6_{0.8}$ | $34.3_{1.4}$ | $34.0_{1.0}$ | $55.0_{2.1}$ | $52.4_{2.6}$ | $54.2_{2.5}$ | $52.8_{3.7}$ |
| ANLI-R2 | 33.4 | $33.6_{0.7}$ | $36.4_{3.3}$ | $33.3_{2.2}$ | $43.5_{0.9}$ | $44.4_{2.2}$ | $45.1_{1.1}$ | $45.1_{1.8}$ |
| ANLI-R3 | 33.5 | $34.2_{0.8}$ | $33.4_{2.0}$ | $33.6_{1.4}$ | $44.6_{0.9}$ | $42.3_{2.5}$ | $45.1_{2.1}$ | $44.5_{1.4}$ |
| CB | 50.0 | $57.5_{0.8}$ | $84.2_{3.2}$ | $88.5_{2.7}$ | $93.2_{3.4}$ | $93.2_{3.6}$ | $96.1_{1.9}$ | $95.7_{0.9}$ |
| Emotion | 35.2 | $42.1_{0.8}$ | $72.2_{2.4}$ | $76.9_{2.4}$ | $69.0_{1.3}$ | $80.1_{2.0}$ | $75.2_{4.3}$ | $80.1_{1.6}$ |
| Enron | 50.9 | $53.3_{0.4}$ | $95.1_{2.3}$ | $96.0_{0.8}$ | $97.1_{0.3}$ | $97.2_{0.9}$ | $97.1_{0.1}$ | $97.8_{0.4}$ |
| Amazon-CF | 0.00 | $0.04_{0.7}$ | $55.7_{4.8}$ | $64.8_{6.3}$ | $29.8_{4.1}$ | $62.8_{4.2}$ | $64.5_{3.7}$ | $66.3_{3.1}$ |
| CR | 64.2 | $88.9_{0.4}$ | $89.3_{1.8}$ | $91.6_{1.0}$ | $94.0_{0.7}$ | $94.3_{0.8}$ | $92.6_{2.1}$ | $92.6_{1.8}$ |
| SST-5 | 26.3 | $38.9_{1.0}$ | $46.1_{1.1}$ | $50.8_{1.3}$ | $52.3_{1.4}$ | $50.0_{3.4}$ | $49.4_{3.2}$ | $45.6_{5.2}$ |
| Average | 41.1 | $47.3_{1.0}$ | $58.6_{2.6}$ | $60.7_{2.5}$ | $66.2_{1.7}$ | $69.6_{2.6}$ | $68.3_{2.6}$ | $69.2_{2.5}$ |

64 samples per class.

Table 8: Results with T0 upstream model. (H): Handcrafted, (A w/o D): Automated Prompts without dataset label candidates, (A): Automated Prompts.

| Dataset | | Answer Choice | | |
| --- | --- | --- | --- | --- |
| | **Dataset** | **template-tailored** | **Topic-Specific** | **Selected** |
| RTE | entailment/not_entailment | Yes / No | scandal / dictator | Yes / No |
| WiC | No / Yes | run / work | force / sentence | No / Yes |
| WSC | No / Yes | good / Yes | bob / peter | No / Yes |
| ANLI-R1 | entailment / neutral / contradiction | Yes / / No | hound / market / presence | Yes /  / No |
| CB | entailment / contradiction / neutral | Yes / No / no | passage / sentence / funny | Yes / No / no |
| Emotion | sadness/joy/love/anger/fear/surprise | " | negative/pos/good/bad/positive | sadness / joy / love / anger / fear /surprise |
| Enron | ham / spam | Business / 5 | enrononline / pricing | enrononline / pricing |
| Amazon-CF | not-counterfactual / counterfactual | positive / negative | fabric / perfect | not-counterfactual/counterfac |
| CR | negative / positive | negative / positive | mp3player / ipod | mp3player / ipod |
| SST-5 | very negative / negative / neutral / positive / very positive | No / negative / <unk> / Yes / positive | filmmaking / genre / scene / documentary / cinematic | No / negative / <unk>/ Yes / positive |

Table 9: Generated and selected answer choices, when using T0 and 32 samples for seed 0.

| Dataset | Rank | Template |
|---|---|---|
| RTE | 1 | {{premise}} Question: {{hypothesis}} |
| | 2 | {{premise}} \n Is that a paraphrase of the following sentence? \n {{hypothesis}}? |
| | 3 | {{premise}} \n Is that paraphrasing the following sentence? \n {{hypothesis}}? |
| | 4 | {{premise}} Question: {{hypothesis}} |
| | 5 | Sentence 1: {{premise}} \n Sentence 2: {{hypothesis}} \n Question: Does Sentence 1 paraphrase Sentence 2? |
| WiC | 1 | Pick one category for the following text. The options are - {{sentence1}}. {{sentence2}} - {{word}} |
| | 2 | {{sentence1}} - {{sentence2}} Given a choice of categories {{word}}, the text refers to which one? |
| | 3 | This is a correct answer to the following word about {{sentence1}}. \n Answer: {{sentence2}} \n Question: {{word}} |
| WSC | 1 | Pick one category for the following text. The options are - {{sentence1}}. {{sentence2}} - {{word}} |
| | 2 | {{sentence1}} - {{sentence2}} Given a choice of categories {{word}}, the text refers to which one? |
| | 3 | This is a correct answer to the following word about {{sentence1}}. \n Answer: {{sentence2}} \n Question: {{word}} |
| ANLI | 1 | {{premise}} Question: {{hypothesis}} |
| | 2 | {{premise}} \n Is that a paraphrase of the following sentence? \n {{hypothesis}}? |
| | 3 | {{premise}} \n Is that paraphrasing the following sentence? \n {{hypothesis}}? |
| | 4 | {{premise}} Question: {{hypothesis}} |
| | 5 | Sentence 1: {{premise}} \n Sentence 2: {{hypothesis}} \n Question: Does Sentence 1 paraphrase Sentence 2? |
| CB | 1 | {{premise}} Question: {{hypothesis}} |
| | 2 | {{premise}} \n Is that a paraphrase of the following sentence? \n {{hypothesis}}? |
| | 3 | {{premise}} \n Is that paraphrasing the following sentence? \n {{hypothesis}}? |
| | 4 | {{premise}} Question: {{hypothesis}} |
| | 5 | Sentence 1: {{premise}} \n Sentence 2: {{hypothesis}} \n Question: Does Sentence 1 paraphrase Sentence 2? |
| Emotion | 1 | {{text}} How does the viewer feel about the movie? |
| | 2 | {{text}} How does the reviewer feel about the movie? |
| | 3 | {{text}} Did I regret it? |
| | 4 | If you ask me whether I like this place? The answer is {{text}} |
| | 5 | {{text}} Overall, the experience is |
| Enron | 1 | {{text}} If you ask me whether I will come again, my answer is |
| | 2 | {{text}} Will you come here again? |
| | 3 | {{text}} What is the sentiment expressed in this text? |
| | 4 | Based on that, my rating for this place is {{text}} |
| | 5 | {{text}} If you ask me whether I like this place? The answer is |
| Amazon-CF | 1 | {{text}} How does the reviewer feel about the movie? |
| | 2 | {{text}} Did the reviewer enjoy the movie? |
| | 3 | {{text}} How does the viewer feel about the movie? |
| | 4 | Based on this review, would the user recommend this product? \n === \n Review: {{text}} \n Answer: |
| | 5 | {{text}} What is the sentiment expressed by the reviewer for the movie? |
| CR | 1 | Based on this review, would the user recommend this product? === \n Review: {{text}} \n Answer: ||| |
| | 2 | {{text}} How does the viewer feel about the movie? |
| | 3 | {{text}} How does the reviewer feel about the movie? |
| | 4 | Review: \n {{text}} \n Overall rating: |
| | 5 | {{text}} \n Overall, the experience is |
| SST-5 | 1 | {{text}} How does the viewer feel about the movie? |
| | 2 | {{text}} How does the reviewer feel about the movie? |
| | 3 | {{text}} What sentiment does the writer express for the movie? |
| | 4 | The following movie review expresses what sentiment? {{text}} |
| | 5 | {{text}} Did the reviewer enjoy the movie? |

Table 10: Retrieved templates, when using T0 and 32 samples for seed 0.