# OpenReview forum: "Automated Few-Shot Classification with Instruction-Finetuned Language Models"
_EMNLP/2023/Conference — EMNLP 2023 Findings_

### Official Review · Reviewer_YSqb · 2023-08-03

**Typos Grammar Style And Presentation Improvements:** No
**Soundness:** 4

**Excitement:**

3: Ambivalent: It has merits (e.g., it reports state-of-the-art results, the idea is nice), but there are key weaknesses (e.g., it describes incremental work), and it can significantly benefit from another round of revision. However, I won't object to accepting it if my co-reviewers champion it.

**Missing References:**

No

**Paper Topic And Main Contributions:**

This paper aims to address the problem of hand-crafted prompt design on few-shot learning. The proposed AuT-few model can automatically generates appropriate prompt in discrete output space for downstream tasks. Extensive empirical studies on 12 datasets are conducted and results shows that the sensitivity of prompt design is different for template and answer choices, arguing the importance of rationally designing prompts in answer choices. Based on the experimental results, the article proposes a Retrieval approach for template design and two approaches, template-tailored and topic-specific, for answer choices. Extensive experiments demonstrate that AuT-Few outperforms the previous SOTA T-Few by 2 points on 8 tasks on 12 datasets, and has good robustness to different models and out-of-domain datasets. The training and inference overheads are also calculated, although it is inferior to SOTA or equivalent models in terms of training cost or inference speed, it is still in the comparable range, requiring only a few hours of training on an A10G single GPU, hence providing new automatic methodological prompt ideas to the few-shots community and easy to fuse on existing architectures.

**Questions For The Authors:**

Question A: Despite the conclusion that “neither template tailored answer choices nor multiple distinct answer choice configurations is needed in the empirical experiments” drawn in Chapter 3, would it be possible to combine the three answer choices, Dataset, template-tailored, and Topic-Specific, in a discrete space rather than choosing the one with the highest score?

Question B: Why does Topic-Specific answer choice only consider single-word to semantically express the underlying content for each answer class, and was there any experiment on word number ablation?

Question C: When comparing the FLOPS of training and reasoning with other models, could you explain in detail which improvements in your setting lead to the increase of FLOPS, and what are the marginal benefits they bring to the increase of FLOPS? Could you compare the training and inference time cost with other benchmark models in more detail based on the marginal benefits?


**Reasons To Accept:**

The article empirically investigates the robustness of two types of prompts, task template and answer choices, on multiple classification task datasets, and proves that template prompt is more robust than answer choices prompt, providing strong empirical and hence a simple retrieve method can be used for template prompts. Answer choices are more sensitive to prompts, so the authors propose two methods, template-tailored and topic-specific answer choices, based on retrieve. answer choices, and proved the rationality of these two methods in ablation experiments.

The authors also empirically verified the feasibility of fusing the two methods of IA3 and LoRA, which provides ideas for the fusion of current main different methods in PEFT. In addition, the authors conduct extensive experimental comparisons with the pre-SOTA T-Few on mainstream models and classification task datasets, and reach SOTA in RAFT benchmark to validate the effectiveness of the proposed methods.


**Reasons To Reject:**

The paper’s main improvement which focuses on prompt setting of task template and answer choices in discrete space relies on the architecture of the pre-SOTA T-few, with fewer innovations in the PEFT modeling framework.

In addition, the reasons for the training time and increased computation are not sufficiently explained when comparing with the former SOTA. In terms of template prompt retrieval, there is a lack of analysis on the impact of the overall quality of collection prompts on template prompts, and subsequent researchers may have difficulties in finding suitable metrics to measure the quality of the prompt collection used to retrieve templates in the face of an ever-expanding prompt collection.

**Reproducibility:**

4: Could mostly reproduce the results, but there may be some variation because of sample variance or minor variations in their interpretation of the protocol or method.

**Reviewer Confidence:**

4: Quite sure. I tried to check the important points carefully. It's unlikely, though conceivable, that I missed something that should affect my ratings.

---

> ### Author Rebuttal · Authors · 2023-08-28
>
> We thank the reviewer for the time and comments on our paper! We address the reviewer's questions and the modifications we make to our paper based on the reviewer's suggestions below:
>
> **Question A: Combination of generated answer choice configurations**
>
> This is a good question. We conducted experiments that combine the three answer choice configurations by: (i) optimizing directly over multiple answer choices $\psi_j(y)$ and running a Monte-Carlo approximation during inference over both templates *and* answer choices, (ii) combining the textual descriptions for each class by string concatenation and/or autoregressive generation (e.g. for RTE, the entailment class would be described as “Yes, entailment or scandal”). However, for both approaches, we observe negative results, since some answer choice configurations are not informative for every dataset but instead can be considered misleading at times (e.g. the topic-specific answer choice “scandal” for RTE). Pre-filtering misleading answer choices to combine exclusively informative choices might lead to more positive results, however,  more experiments are difficult to conduct given the time constraint of the rebuttal but we will report additional findings in the paper.
>
> **Question B: Topic-specific answer choices with descriptions spanning multiple words**
>
> We ran preliminary experiments concatenating the top-k ranked words for a single class together (for $k \in \\{1, 2, 3\\}$), with $k=1$ producing the best scores. More generally, we observed a tendency for the LMs to prefer shorter answer choices across answer choice configurations. We will conduct a more detailed analysis of the quantitative characteristics of the selected answer choices and we will report our additional findings in the additional space if accepted.
>
>
> **Question C: Explanation and breakdown of increased computation**
>
> We acknowledge that our analysis of the computational costs would benefit from a breakdown into how individual components impact both runtime and overall performance. We will address this in the camera-ready version of the paper if accepted.
>
> Runtime during inference: AuT-Few’s Monte-Carlo approximation over all retrieved templates is the sole cause for the computational increase, increasing compute by a factor of 5 (since 5 templates are retrieved). Running AuT-Few without Monte-Carlo approximation (i.e. random sampling of templates) makes AuT-Few *as efficient* as T-Few during inference while still outperforming it, see Table 4 in our ablation study. Moreover, AuT-Few with the smaller BART0 backbone requires only 60% of SetFit’s computation during inference without Monte-Carlo approximation, yet scores higher. As also noted to Reviewer 1, we acknowledge that the use of Monte-Carlo approximation during inference distracts from the overall efficiency of our approach and we will ensure to clarify this point by moving the scores with Monte-Carlo approximation to the appendix if accepted, making the efficiency analysis easier to follow.
>
> Costs for training: Training is almost exclusively impacted by running 3-fold cross-validation for each answer choice configuration to select the most appropriate one (the additional parameters added by LoRA are negligible in comparison). As seen in the ablation study (Table 4), the selection mechanism results in an average improvement of 0.8 points over the next base configuration (dataset label text) while also reducing variance. When not assuming the availability of an intelligible dataset label text (frequently the case with Hugginface datasets), the average improvement is 2.1 points (AuT-Few scores 65.4 versus 63.3 with template-tailored choices). Since training times in a few-shot scenario are generally very short (magnitudes shorter than for smaller models like BERT in a fully supervised setting), we do not consider these additions to the runtime problematic.

---

### Official Review · Reviewer_rdNw · 2023-08-05

**Soundness:** 3

**Excitement:**

3: Ambivalent: It has merits (e.g., it reports state-of-the-art results, the idea is nice), but there are key weaknesses (e.g., it describes incremental work), and it can significantly benefit from another round of revision. However, I won't object to accepting it if my co-reviewers champion it.

**Paper Topic And Main Contributions:**

The paper introduces AuT-Few, a method that eliminates the need for handcrafted prompts in few-shot learning. AuT-Few uses a retrieval module to fetch task instructions and generates two distinct class descriptions, selecting the best via cross-validation. Across 12 datasets and 8 tasks, AuT-Few outperforms leading few-shot methods and ranks top on the RAFT benchmark, all without task-specific prompts for new tasks.

**Questions For The Authors:**

Please refer to [reasons to reject]

**Reasons To Accept:**

1. The paper proposes AuT-Few which could replace hand-designed task-specific prompts which can remove human intervention and expertise.

2. Paper is well-written

**Reasons To Reject:**

1. Although the method outperforms T-Few on average, it only wins 4 out of 12 datasets (Table 1). Thus I think it is an overstatement that it outperforms T-Few.

2. What does inverted handcrafted answer choices mean (line 259)? Could you clarify?

**Reproducibility:**

5: Could easily reproduce the results.

**Reviewer Confidence:**

3: Pretty sure, but there's a chance I missed something. Although I have a good feel for this area in general, I did not carefully check the paper's details, e.g., the math, experimental design, or novelty.

---

> ### Author Rebuttal · Authors · 2023-08-28
>
> We would like to thank the reviewer for the time and comments on our paper! We address the reviewer's questions and modifications we make to our paper based on the reviewer's suggestions below:
>
> **Although the method outperforms T-Few on average, it only wins 4 out of 12 datasets (Table 1). Thus I think it is an overstatement that it outperforms T-Few.**
>
> We acknowledge the reviewer’s point and will explicitly state the win rate of AuT-Few in the main results of the paper. However, we would like to add that while AuT-Few wins in 4 out of 12 datasets, the delta in performance for the cases where it wins is much larger than for the ones it loses (except to WSC), arguing in favour of AuT-Few’s robustness and overall performance. This observation is also confirmed on the RAFT benchmark, where AuT-Few performs more consistently than all other approaches, having the best average rank.
>
> **Q: What does inverted handcrafted answer choices mean (line 259)? Could you clarify?**
>
> We will clarify the definition in section 3 of the paper. We recognize that *inverted* answer choices should rather be called *reversed* answer choices to avoid confusion and we will adjust the term in the paper. This prompt modification reverses the mapping from the labels to the answer choices, i.e. if $\psi(0) = \text{No}$, $\psi(1) = \text{Yes}$, then $\text{reversed}(\psi)(0) = \text{Yes}$, and $\text{reversed}(\psi)(1) = \text{No}$. For cases with more than two classes: e.g. $\psi(0) = \text{sadness}$, $\psi(1) = \text{joy}$, $\psi(2) = \text{love}$, $\psi(3) = \text{anger}$, $\psi(4) = \text{fear}$, $\psi(5) = \text{surprise}$, then $\text{reversed}(\psi)(0) = \text{surprise}$, $\text{reversed}(\psi)(1) = \text{fear}$,  $\text{reversed}(\psi)(2) = \text{anger}$, $, \cdots, $ $\text{reversed}(\psi)(5) = \text{sadness}$. Hence, reversed answer choices are explicitly destructive prompts and they subsequently have the highest negative impact on performance (see Figure 3).

---

### Official Review · Reviewer_YfFv · 2023-08-05

**Soundness:** 3

**Excitement:**

3: Ambivalent: It has merits (e.g., it reports state-of-the-art results, the idea is nice), but there are key weaknesses (e.g., it describes incremental work), and it can significantly benefit from another round of revision. However, I won't object to accepting it if my co-reviewers champion it.

**Paper Topic And Main Contributions:**

The study presents an innovative method for generating prompts automatically, eliminating the need for manual crafting. This is achieved by retrieving appropriate task instructions from an instruction-tuning knowledge base and answer choices. Notably, this process can be executed with access only to training samples and categorical labels.
Their method, known as AuT-Few, is tested across 12 datasets and is shown to outperform the current state-of-the-art methods in few-shot learning. Further demonstrating its efficacy, the authors highlight the generalizability of AuT-Few by reporting successful results on tasks that were not seen during the training.

**Questions For The Authors:**

please refer to the reasons to reject section.

**Reasons To Accept:**

1. Autohrs conduct an analysis of diverse types of prompts, comparing their performance across a range of datasets. This comprehensive comparison can provide valuable insights that may greatly benefit future research in this field.

2. Proposed method eliminates the need for manually hand-crafted prompts or any prerequisite background knowledge for training new tasks.



**Reasons To Reject:**

1. You've mentioned in lines 009-012 that the instruction fine-tuned Language Model (LM) demonstrates significant prompt robustness. Given this, could you explain why there's a need for automatic prompt generation rather than using general prompts?

2. It is noted that the inference time for this model is significantly longer than that of previous works. This aspect seems to detract from the model's overall effectiveness.

3. The appendix includes examples of some strange automated choices that, despite their oddity, result in good performance. Could you provide an analysis explaining this?

**Reproducibility:**

4: Could mostly reproduce the results, but there may be some variation because of sample variance or minor variations in their interpretation of the protocol or method.

**Reviewer Confidence:**

4: Quite sure. I tried to check the important points carefully. It's unlikely, though conceivable, that I missed something that should affect my ratings.

---

> ### Author Rebuttal · Authors · 2023-08-28
>
> We thank the reviewer for the time and comments on our paper! We address the reviewer's questions and the modifications we make to our paper based on the reviewer's suggestions below:
>
> **Q: [...] Given this, could you explain why there's a need for automatic prompt generation rather than using general prompts?**
>
> We acknowledge that  “prompt robustness” in the abstract as a description for our findings in section 3 is too vague and potentially leaves the reader confused regarding our subsequent motivation for AuT-Few. We will ensure to adjust the abstract accordingly.
> Instruction fine-tuned language models in the context of PEFT exhibit surprisingly little variability (but not none!) on some dimensions of its design space (i.e. task-specific templates, template-tailored answer choices, multiple answer-choice configurations), however, they are susceptible to some others (i.e. answer choice configuration). Hence, an LM’s prompt remains a relevant hyperparameter worth optimizing, either by humans or automatically. For some datasets, the impact of optimizing the prompts is larger than for others (e.g. on Amazon Counterfactual, WSC, RTE, or ANLI prompts have a substantial impact on performance while on e.g. CR the performance is virtually unaffected by any modifications).
>
> AuT-Few is designed around the observations made in section 3: (i) distribute computational efforts for prompt optimization to reflect the components of the prompts that the instruction-tuned LM is more susceptibility to, i.e. spend more resources on answer choice automation than on template automation (template retrieval is extremely efficient but still provides performance improvements) (ii) optimize templates independently from answer choices for efficiency reasons (contrary to e.g. Gao et al., 2021 [1]), since template-tailored answer choices are not needed, (iii) select the single-best answer choice configuration instead of trying to find multiple distinct ones, as commonly done with handcrafted prompts.
>
> **It is noted that the inference time for this model is significantly longer than that of previous works. [...]**
>
> For clarification: the inference time for AuT-Few is only longer when using Monte-Carlo approximation over all retrieved templates, however, without it (i.e. random sampling of retrieved templates) AuT-Few is *as efficient* as T-Few during inference while still outperforming it, see Table 4 in our ablation study. Notably, AuT-Few with the smaller BART0 backbone requires only 60% of SetFit’s computation during inference without Monte-Carlo, yet scores higher. Considering the efficiency of SetFit, we believe this is a notable achievement. We agree with the reviewer that the use of Monte-Carlo approximation during inference distracts from the overall efficiency of our model and will ensure to clarify this point by moving the scores with Monte-Carlo approximation to the appendix if accepted.
>
> **Q: The appendix includes examples of some strange automated choices [...] Could you provide an analysis explaining this?**
>
> One surprising finding in section 3 is that on a couple of datasets (CR and Enron) completely uninformed and even destructive answer choices (i.e. inverted answer choices, see our comment to Reviewer 2 for a clear definition) have a negligible impact on the performance of instruction-tuned LMs in the context of PEFT. Both datasets are virtually unaffected by any modifications we made to both task templates and the answer choices (see Figure 3).  For CR, AuT-Few’s cross-validation strategy for selecting the best answer choice subsequently measures almost identical scores for all three choice configurations (90.1, 89.8, 90.4 for the dataset, template-tailored, and topic-specific choices, respectively), resulting in the seemingly erroneously selection of the topic-specific answer choices (mp3player / ipod). We will add a qualitative discussion of the selected answer choices and templates and analyse potential causes for our observations in section 3 regarding CR and Enron using the additional space provided in the camera-ready version if accepted.
>
> [1] https://aclanthology.org/2021.acl-long.295.pdf

---

### Meta-Review · Area_Chair_bVPW · 2023-09-11

**Recommendation:** 4

**Metareview:**

This paper deals with automated prompt construction for few-shot (parameter-efficient fine-tuning, not in-context) learning for instruction-tuned LLMs. The authors first demonstrate that instruction-tuned LLMs are pretty robust to the selection of instruction prompts for classification task, which then informs an automated approach to prompt selection, based on prompt retrieval (over a catalog of instructions on which the backbone model was initially (pre)trained) and strategy for generating candidates for answer choices (i.e., class lexicalizations), based on template-conditioned probabilities of tokens for a class (on the training data). Evaluation of Aut-Few on a dozen short-text classification datasets renders it effective (more on reviewer concerns raised on Aut-Few performance below).

There is some logical discrepancy between the initial study, which shows that instruction-tuned LLMs are robust to the choice of the prompt, and the rest of the paper that aims to automate prompt (task-description) selection/creation. The authors explain (in the rebuttal) that, despite the fact that instruction-tuned LLMs exhibit more robustness w.r.t. choice of task description than vanilla LLMs (i.e., not instruction-tuned), there is still variation, especially w.r.t. lexicalizations of classes. The finding of robustness of instruction-based LLMs is indeed somewhat at odds with the motivation for proposing Aut-Few, and the authors should address this in the structure of the paper.

The main concern raised in the reviews is that of performance of Aut-Few in comparison to T-Few, which manually selects the prompts. The concern raised was that Aut-Few outperforms T-Few only on a minority of the evaluation datasets. The authors clarify the significance of their results in the rebuttal: on 3 datasets, Aut-Few is significantly better than T-Few, whereas on 1 dataset the opposite is true; on the remaining datasets the difference in performance between the two is not statistically significant. The crucial point here, that the reviewer may have missed, however, is that T-Few implies careful manual selection of the prompt, whereas Aut-Few removes the need for any manual effort.

Although one could argue that neither the retrieval step nor class label generation component are greatly innovative -- i.e., they can be seen as relatively straightforward extensions of existing body of work on prompting LLMs -- they are meaningful and convincing. Moreover, the practical benefit of this work -- removal of the need for manual effort in prompt design for instruction-tuned LLMs -- is substantial.

---

### Decision · Program_Chairs · 2023-10-07

**Decision:**

Accept-Findings

**Comment:**

This paper deals with automated prompt construction for few-shot (parameter-efficient fine-tuning, not in-context) learning for instruction-tuned LLMs. The authors first demonstrate that instruction-tuned LLMs are pretty robust to the selection of instruction prompts for classification task, which then informs an automated approach to prompt selection, based on prompt retrieval (over a catalog of instructions on which the backbone model was initially (pre)trained) and strategy for generating candidates for answer choices (i.e., class lexicalizations), based on template-conditioned probabilities of tokens for a class (on the training data). Evaluation of Aut-Few on a dozen short-text classification datasets renders it effective (more on reviewer concerns raised on Aut-Few performance below).

There is some logical discrepancy between the initial study, which shows that instruction-tuned LLMs are robust to the choice of the prompt, and the rest of the paper that aims to automate prompt (task-description) selection/creation. The authors explain (in the rebuttal) that, despite the fact that instruction-tuned LLMs exhibit more robustness w.r.t. choice of task description than vanilla LLMs (i.e., not instruction-tuned), there is still variation, especially w.r.t. lexicalizations of classes. The finding of robustness of instruction-based LLMs is indeed somewhat at odds with the motivation for proposing Aut-Few, and the authors should address this in the structure of the paper.

The main concern raised in the reviews is that of performance of Aut-Few in comparison to T-Few, which manually selects the prompts. The concern raised was that Aut-Few outperforms T-Few only on a minority of the evaluation datasets. The authors clarify the significance of their results in the rebuttal: on 3 datasets, Aut-Few is significantly better than T-Few, whereas on 1 dataset the opposite is true; on the remaining datasets the difference in performance between the two is not statistically significant. The crucial point here, that the reviewer may have missed, however, is that T-Few implies careful manual selection of the prompt, whereas Aut-Few removes the need for any manual effort.

Although one could argue that neither the retrieval step nor class label generation component are greatly innovative -- i.e., they can be seen as relatively straightforward extensions of existing body of work on prompting LLMs -- they are meaningful and convincing. Moreover, the practical benefit of this work -- removal of the need for manual effort in prompt design for instruction-tuned LLMs -- is substantial.